# Regulation of protein complex partners as a compensatory mechanism in aneuploid tumors

Gökçe Senger[1], Stefano Santaguida[1,2], Martin H Schaefer[1]*

[1]Department of Experimental Oncology, IEO European Institute of Oncology IRCCS, Milan, Italy; [2]Department of Oncology and Hemato-Oncology, University of Milan, Milan, Italy

**Abstract** Aneuploidy, a state of chromosome imbalance, is a hallmark of human tumors, but its role in cancer still remains to be fully elucidated. To understand the consequences of whole-chromosome-level aneuploidies on the proteome, we integrated aneuploidy, transcriptomic, and proteomic data from hundreds of The Cancer Genome Atlas/Clinical Proteomic Tumor Analysis Consortium tumor samples. We found a surprisingly large number of expression changes happened on other, non-aneuploid chromosomes. Moreover, we identified an association between those changes and co-complex members of proteins from aneuploid chromosomes. This co-abundance association is tightly regulated for aggregation-prone aneuploid proteins and those involved in a smaller number of complexes. On the other hand, we observed that complexes of the cellular core machinery are under functional selection to maintain their stoichiometric balance in aneuploid tumors. Ultimately, we provide evidence that those compensatory and functional maintenance mechanisms are established through post-translational control, and that the degree of success of a tumor to deal with aneuploidy-induced stoichiometric imbalance impacts the activation of cellular protein degradation programs and patient survival.

*For correspondence:
martin.schaefer@ieo.it

**Competing interest:** The authors declare that no competing interests exist.

## Editor's evaluation

This paper will be of interest to the cancer biology community. The study leverages high-throughput genomic and proteomic data to evaluate the role of aneuploidy on functional pathway changes in cancer.

## Introduction

Aneuploidy, whole-chromosome- or chromosome-arm-level alterations, affects expression of a large number of genes simultaneously - most directly by providing additional or diminished copies of genes in - or decreasing their transcription. Typically, aneuploidy is detrimental for normal cells, among other reasons as it causes stoichiometric imbalances in protein complexes involving proteins encoded on the aneuploid chromosome (*Brennan et al., 2019*; *Santaguida and Amon, 2015a*). However, around 90% of solid tumors have aneuploid karyotypes (*Ben-David et al., 2019*; *Taylor et al., 2018*), raising the question of how cancer cells can tolerate the massive amount of transcriptomic and proteomic changes.

Previous studies demonstrated correlated expression between gene copy number and transcriptome level for genes on aneuploid chromosomes while there is a buffering effect at the proteome level adjusting protein levels, especially for protein complex subunits (*Stingele et al., 2012*; *Dephoure et al., 2014*). This post-transcriptional compensatory mechanism preventing excess translation

of complex members has also been characterized for copy number alterations (CNAs) in different contexts such as yeast (*Ishikawa et al., 2017*) and cancer (*Gonçalves et al., 2017*). Structural properties of proteins have an effect on the degree of post-transcriptional buffering on changes induced by CNAs, for example, proteins with larger interface size showed larger degree of buffering (*Sousa et al., 2019*). Recently, post-translational regulation has also been identified as a dosage compensation mechanism in response to aneuploidy in cancer cell lines (*Schukken and Sheltzer, 2021*).

Transcriptome analysis revealed that aneuploidy largely affects expression of genes on other chromosomes too (*Nawata et al., 2011*; *Upender et al., 2004*). For CNAs, a correlated increase in the abundance of co-complex members encoded by genes outside the copy number amplified region has been shown (*Gonçalves et al., 2017*) raising the question of whether this could contribute to the expression changes in aneuploid cells even on diploid chromosomes. Together these results suggest the importance of compensation mechanisms to buffer differentially expressed transcripts in response to the amplification or loss of genomic regions and in particular to mitigate stoichiometric imbalances

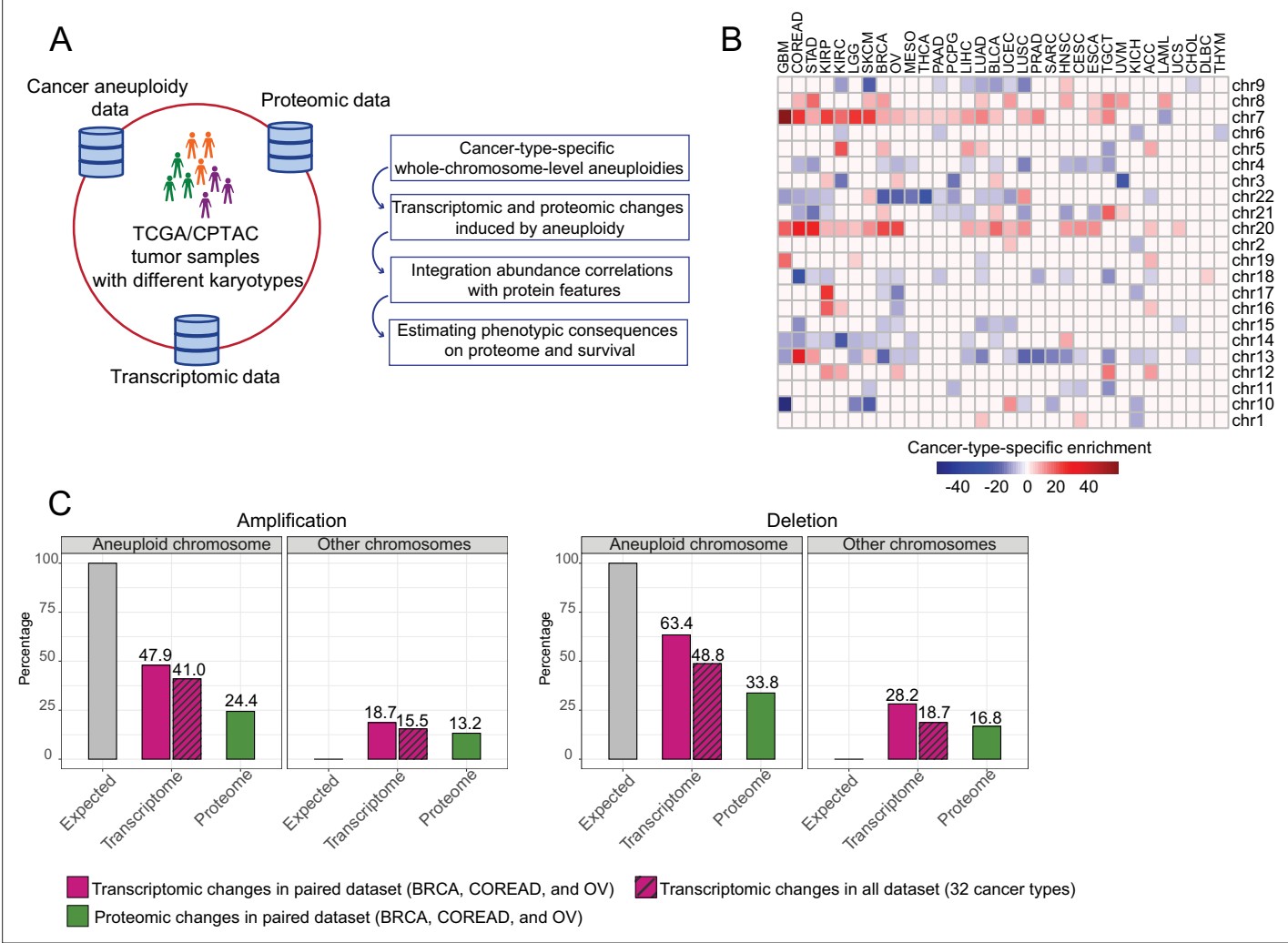

**Figure 1.** Transcriptomic and proteomic changes in aneuploid tumors. (**A**) Data used in this study and schematic representation of the performed analyses. (**B**) Cancer-type-specific, whole-chromosome-level alterations across 32 cancer types. The color encodes the degree of their enrichment (standard residuals of the chi-square test multiplied by the alteration score [–1 in the case of deletion and 1 in the case of amplifications]). (**C**) Average percentage of differentially expressed genes or abundant proteins on aneuploid and other, non-aneuploid chromosomes (among the detected genes on the respective chromosomes).

The online version of this article includes the following figure supplement(s) for figure 1:

**Figure supplement 1.** Proteome data coverage and differential expression changes on other chromosomes.

in protein complexes. However, we still lack a global understanding on the effects of aneuploidy on the expression of genes on other, non-aneuploid chromosomes in a cancer context.

Here, we study transcriptomic and proteomic changes induced by aneuploidy in cancer patients (*Figure 1A*) and extend the scope of previous studies by focusing on expression and abundance changes on other, non-aneuploid chromosomes. We show that protein complex subunits of other chromosomes tend to maintain their abundance levels unless they form a complex with differentially abundant proteins encoded by genes located on the aneuploid chromosome. We further demonstrate that this co-abundance regulation is dependent on aggregation propensity and promiscuity of the aneuploid complex partners and controlled by post-translational mechanisms. Our findings highlight a complementary mechanism acting to deal with the excess amount of expression changes induced by aneuploidy: Coordinated abundance changes of complex partners might prevent aggregation of unpaired complex members.

## Results

### Widespread transcriptome and proteome deregulation in aneuploid tumors

To study the effect of whole chromosomal alterations on cancer transcriptomes and proteomes, we first identified cancer-type-specific, whole-chromosome-level amplifications and deletions that occurred at higher frequencies than would be expected by chance in 10,522 samples analyzed in The Cancer Genome Atlas (TCGA) by using a previously established cancer aneuploidy estimate (*Taylor et al., 2018*). In total, we detected 203 whole-chromosome-level alterations including 86 amplifications and 117 deletions for 32 cancer types (*Figure 1B*, *Supplementary file 1*). Then, for each detected aneuploidy case, we split the set of samples into those containing the respective chromosome number aberration and those diploid for the respective chromosome and then tested for differential expression of all genes between the sets. We found that on average 41% and 48% of the genes located on amplified and deleted chromosomes, respectively, changed expression (*Figure 1C*). Besides those intuitively expected gene expression changes on the aneuploid chromosomes, we observed a surprisingly large number of expression changes happening on other, typically diploid chromosomes (15 and 18% of genes on average for amplification and deletion cases, respectively; *Figure 1C*).

We observed that often chromosomes tend to be co-amplified. We therefore tested for statistical dependence between amplification events of chromosomes. For the 86 cancer-type-specific amplifications, we identified 305 co-amplifications that occurred more frequently than expected by chance covering 60 out of 86 cancer-type-specific amplifications (adjusted p-value <0.01, chi-square test; *Supplementary file 1*). We wondered if this could explain the relatively high number of differentially expressed genes on other chromosomes. To test this, we quantified the contribution of each chromosome to the transcriptional dysregulation by dividing the number of differentially expressed genes from each chromosome to the total number of genes on that chromosome. We then compared the average contribution of co-amplified chromosomes to that of non-co-amplified chromosomes across the 60 cancer-type-specific amplifications. We found that there is no significant difference between the medians of these chromosome groups (p=0.4, paired Wilcoxon test; *Figure 1—figure supplement 1B, C*) suggesting that they do not substantially contribute to the overall transcriptional dysregulation on other chromosomes.

To further understand the effect of those expression changes in response to aneuploidy on the proteome, we collected corresponding proteome abundance data, which is available from the Clinical Proteomic Tumor Analysis Consortium (CPTAC) for 298 TCGA tumor samples comprising breast (BRCA) (*Cancer Genome Atlas Network, 2012*; *Mertins et al., 2016*), ovarian (OV) (*Cancer Genome Atlas Research Network, 2011*; *Zhang et al., 2016*), and colorectal adenocarcinoma (COREAD) (*The Cancer Genome Atlas Network, 2012*; *the NCI CPTAC et al., 2014*) cancer types. Then, for 13 and 20 cancer-type-specific, whole-chromosome-level amplifications and deletions, respectively (found in above mentioned 3 cancer types), we detected protein abundance changes between aneuploid samples (with the amplified/deleted chromosome) and diploid samples. We observed that a smaller number of proteins showed abundance changes compared to our observations at the transcriptome level (after normalizing for the largely different gene coverage between the transcriptomics and proteomics datasets) both for amplification and deletion cases (*Figure 1C*).

One of the major consequences of aneuploidy is its impact on cell proliferation (***Santaguida and Amon, 2015a***). We, therefore, investigated the molecular pathways associated with frequently dysregulated genes and proteins of other chromosomes in aneuploid tumors. Indeed, we found gene ontology (GO) terms belonging to cell cycle and cell cycle processes to be among the top dysregulated gene sets (***Supplementary file 2***).

Together these observations suggest that attenuation mechanisms are in place, and we observe regulation both on transcription and translation level which prevent that all genes located on the aneuploid chromosomes are deregulated. This effect is stronger on protein than on RNA level indicating that on top of regulation of transcription, translation control or protein degradation might play a role. At the same time, we observed a surprisingly large number of dysregulation events on chromosomes other than the aneuploid one raising the question of the purpose of the up- and down-regulation of hundreds of genes in response to specific aneuploidies.

## Complex members tend to be co-deregulated

To test if the vast changes on the cellular proteome in aneuploid cells outside the aneuploid chromosome could be explained by compensation mechanisms for changes in complex stoichiometry

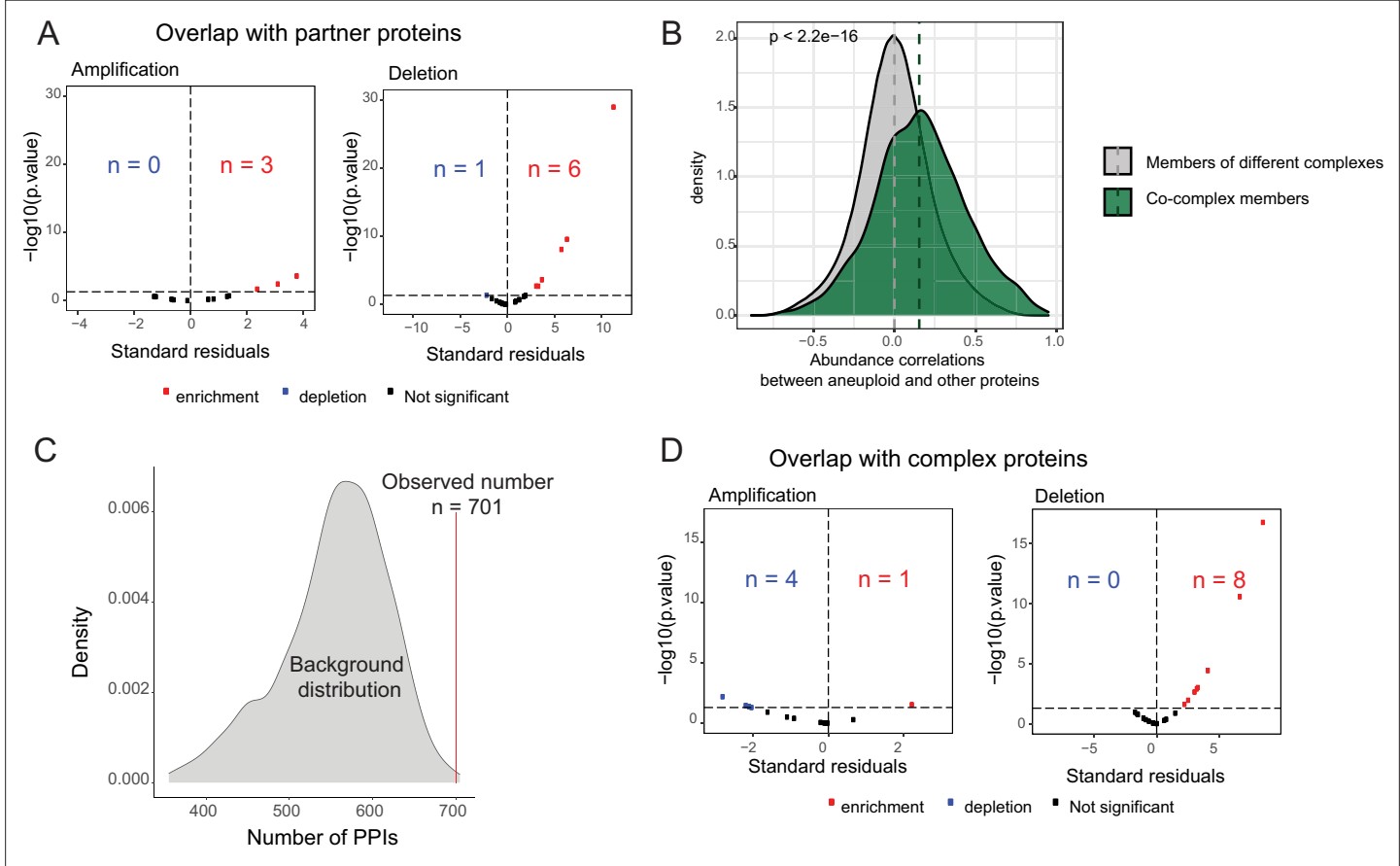

**Figure 2.** Enrichment of partners of aneuploid proteins in differentially abundant proteins on other chromosomes. (**A**) Standard residuals and p-values for the overlap between co-complex members of differentially abundant proteins on aneuploid chromosomes and differentially abundant proteins on other chromosomes for 13 amplifications and 20 deletions. (**B**) Protein abundance correlations between differentially abundant proteins on aneuploid chromosomes and their co-complex and non-complex subunits. Correlations were calculated across cancer samples, separately for each cancer type, and then pooled. Wilcoxon test was used to determine whether two distributions are significantly different. (**C**) The number of protein-protein interactions (PPIs; n=701) between differentially abundant proteins of aneuploid chromosomes and those on other chromosomes against the background distribution for COREAD chromosome 7 amplification. (**D**) Standard residuals and p-values for the overlap between CORUM complex subunits and differentially abundant proteins on other chromosomes for 13 amplifications and 20 deletions.

The online version of this article includes the following figure supplement(s) for figure 2:

**Figure supplement 1.** Transcriptome-level changes on other chromosomes.

induced by aneuploidy, we performed an association test between co-complex members of differentially abundant proteins encoded on aneuploid chromosomes and differentially abundant proteins encoded on other chromosomes by using human complex information from the mammalian protein complex database CORUM (*Giurgiu et al., 2019*). We observed a general tendency for the differentially abundant proteins of other chromosomes to be complex partners of differentially abundant proteins from the aneuploid chromosome for both whole-chromosome-level amplifications and deletions (p<0.05, chi-square test; *Figure 2A*). We found a moderate percentage (in average 4.47%; *Supplementary file 3*) of differentially abundant proteins on other chromosomes being partners of those on aneuploid proteins. However, the coverage of proteins with CORUM complex information is rather limited (22% of proteins form part of at least one complex in CORUM). When only proteins participating in at least one complex were considered, the average fraction of partner proteins among differentially abundant proteins increased to 12.61% (*Supplementary file 3*). Moreover, comparing protein abundance correlations between differentially abundant proteins on aneuploid chromosomes and their co-complex members with non-complex members showed significantly stronger correlations between proteins of same complexes (p<2.2e-16, Wilcoxon test; *Figure 2B*). Those observations are in line with previous findings claiming that complex organization shapes protein abundance changes in response to CNAs (*Sousa et al., 2019*).

To investigate whether our observations can be generalized to binary protein-protein interactions (PPIs), we used PPI data from the human PPI database HIPPIE (v2.2) (*Alanis-Lobato et al., 2017*) to test if the number of interactions between differentially abundant proteins encoded on the aneuploid and those on other chromosomes is higher than expected by chance. Indeed, we found an enrichment of interactions between those protein sets in 9 out of 13 cancer-type-specific amplifications and 8 of the 20 deletions (p<0.05, randomization test; *Figure 2C*, *Supplementary file 4*). Given the higher coverage of PPI data, we asked again which percentage of differentially abundant proteins on other chromosomes could be potentially explained by their interactions with complex members on aneuploid chromosomes. We found that on average 27.5% of the differentially abundant proteins on other chromosomes interact with those on the aneuploid chromosomes (*Supplementary file 4*). For example, for chromosome 7 in COREAD and chromosome 12 in OV, more than 40%, and for chromosome 5 in BRCA, more than 30% of the differentially abundant proteins interacted with proteins on the amplified chromosomes.

We hypothesized that these abundance changes should only affect co-complex members of differentially abundant proteins on aneuploid chromosomes, but that non-partner complex members should maintain their abundance level to prevent stoichiometric imbalances. To test this, we performed an association test for the overlap between differentially abundant proteins on other chromosomes and all known human complex members curated from CORUM. As a result of this, we found a significant depletion of complex subunits in differentially abundant proteins on other chromosomes for amplification cases (p<0.05, chi-square test; *Figure 2D*). This suggests that complex members overall are stably expressed to prevent disruption of complex stoichiometry upon chromosomal amplification. This effect was not observed in the case of chromosomal deletions in which differentially abundant proteins of other chromosomes are significantly enriched in complex proteins (p<0.05, chi-square test; *Figure 2D*).

In contrast to our observations at the proteome level where we observed a consistent pattern of enrichment for co-regulation of co-complex members, we observed both strong enrichments and depletions of co-complex members of proteins encoded by differentially expressed genes on aneuploid chromosomes in the differentially expressed genes on other chromosomes (adjusted p-value <0.01, chi-square test; *Figure 2—figure supplement 1A*). In addition, we observed a significant enrichment of protein complex subunits among the differentially expressed genes on other chromosomes for 19 out of 28, and 20 out of 31 significant associations for amplification and deletion cases, respectively, at the transcriptome level (adjusted p-value <0.01, chi-square test; *Figure 2—figure supplement 1B*). The lack of consistency for co-regulation of co-complex members at the transcriptome level suggests post-transcriptional compensatory mechanisms to control abundance changes induced by aneuploidy.

## Epigenetic and transcriptional control cannot fully explain the dysregulation on other chromosomes

Previous studies have revealed the role of epigenetic and transcriptional regulatory mechanisms in cancer. Differential DNA methylation and dysregulation of transcription factors (TFs) mediate aberrant

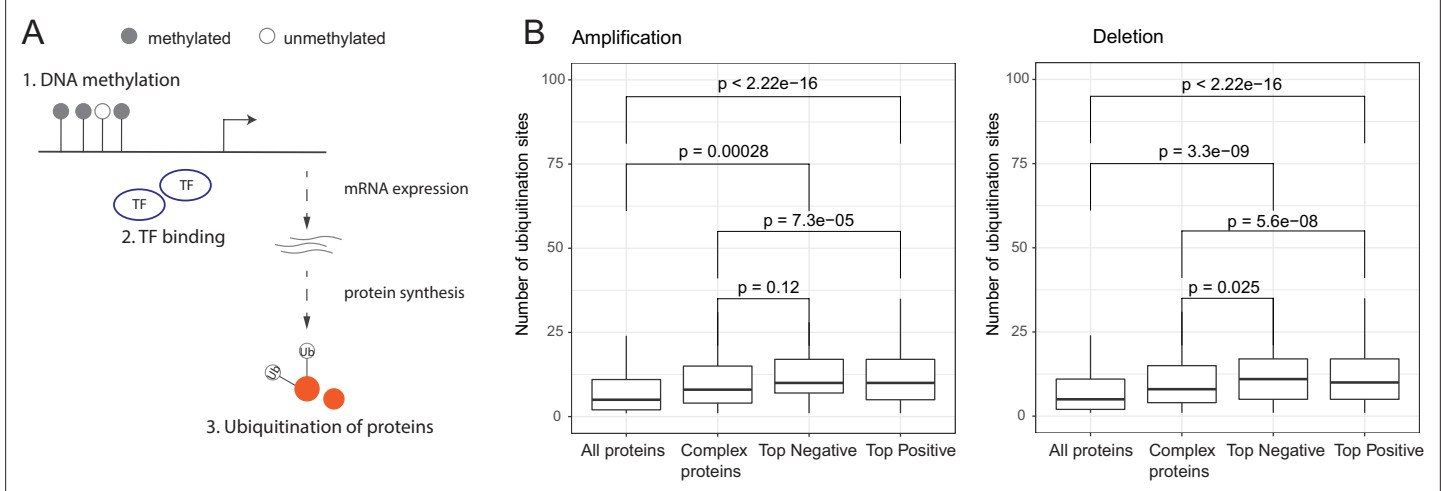

**Figure 3.** Post-translational regulation of co-complex members of aneuploid proteins. (**A**) Overall representation of different levels of gene regulation. (**B**) Number of ubiquitination sites of all, human complex, and top positively and negatively correlated proteins. Wilcoxon test was used to test differences between groups. TF, transcription factors.

The online version of this article includes the following figure supplement(s) for figure 3:

**Figure supplement 1.** Expression changes on other chromosomes cannot be fully explained by transcriptional regulation.

**Figure supplement 2.** Transcriptional regulation on expression changes between tumor and normal, and of co-complex members of aneuploid proteins in aneuploid tumors.

gene expression in cancer (*Baylin and Herman, 2000*; *Ehrlich, 2002*; *Bushweller, 2019*). Thus, we further aimed to disentangle the different regulatory layers underlying expression changes (*Figure 3A*) on other chromosomes induced by aneuploidy. We first tested if those expression changes could be explained by epigenetic silencing via differential DNA methylation of the promoters of genes changing expression in aneuploid samples. Therefore, we compared average DNA methylation levels of differentially expressed genes in aneuploid samples to diploid samples, separately for up- and downregulated genes. We found that downregulated genes are significantly related to higher methylation levels in aneuploid samples in only 6 amplification cases out of 86 (~7%) and in 5 deletion cases out of 117 (~4%; *Figure 3—figure supplement 1A*). We observed significant associations between lower methylation level and upregulated genes in aneuploid samples for few cases (10 out of 86 amplification cases and 16 out of 117 deletion cases) (*Figure 3—figure supplement 1B*). This suggests that epigenetic regulation does not have a substantial contribution to the described genome-wide changes in gene expression induced by aneuploidy.

We then asked if differential expression of TFs on the aneuploid chromosome could explain the large transcriptional changes on other chromosomes. We therefore tested for a large list of ENCODE gene-TF associations if there is an enrichment of targets of differentially expressed TFs on the aneuploid chromosome among differentially expressed genes on the other chromosomes. Performing a randomization test did not reveal an excess of targets for any of the tested, cancer-type-specific chromosomes (*Figure 3—figure supplement 1C*).

As a control, we computed the differentially expressed genes between tumor and healthy samples for 21 TCGA cancer types, where we have tumor vs normal samples. In those downregulated genes are often hypermethylated in cancer (*Figure 3—figure supplement 2A*) suggesting that DNA methylation plays an important role in regulating gene expression during carcinogenesis. Even though not significant we observed a higher number of targets of differentially expressed TFs among differentially expressed genes for 76% of cancer types (*Figure 3—figure supplement 2B*) as compared to 62% in aneuploid tumors. In addition, we observed a higher absolute difference between the number of observed and expected targets in tumor vs normal samples (353.71 and 663.34, respectively, for aneuploid tumors and tumor vs normal). Lastly, we tested the regulatory impact of the differential expression of the well-known cancer-related TF MYC on the expression of its target genes. We found MYC differentially expressed in 14 out of 21 cancer types, and in those cancer types its targets

are significantly enriched among differentially expressed genes (p<0.05, chi-square test). Together these observations show that our measures of transcriptional regulation can capture some regulatory activity in cancer but the absence of signals in aneuploid tumors suggests that transcriptional regulation cannot fully explain the expression changes on other chromosomes.

## Post-translational regulation of partner co-abundance

We observed a stronger association between complex partner co-regulation on proteome as compared to transcriptome level (*Figure 2A*; *Figure 2—figure supplement 1A*) suggesting a central role for translational or post-translational regulation in maintaining complex protein abundance balance in aneuploid cells. To further validate this, we looked at the transcript levels of co-complex members of aneuploid proteins and asked if the corresponding changes could be explained by differential methylation or differential activation of TFs encoded on aneuploid chromosomes. Indeed, we did not observe an overall significant association further supporting the role of translational or post-translational mechanisms on co-abundance regulation (*Figure 3—figure supplement 2C, D*).

Previous studies have suggested that ubiquitination at multiple sites is an efficient signal for degradation (*Dimova et al., 2012*) and further increase in the number of ubiquitination sites is related to higher binding affinity between protein and proteasome (*Lu et al., 2015*). We therefore hypothesized that post-translational ubiquitination of proteins could regulate co-abundance changes of partners of aneuploid proteins on other chromosomes. To test this, we retrieved ubiquitination data from PhosphoSitePlus (*Hornbeck et al., 2015*) and tested if top correlated co-complex members of aneuploid proteins tend to have higher number of ubiquitination sites (as a proxy to identify proteins that can be more easily targeted for degradation). Indeed we found that top correlated partners have

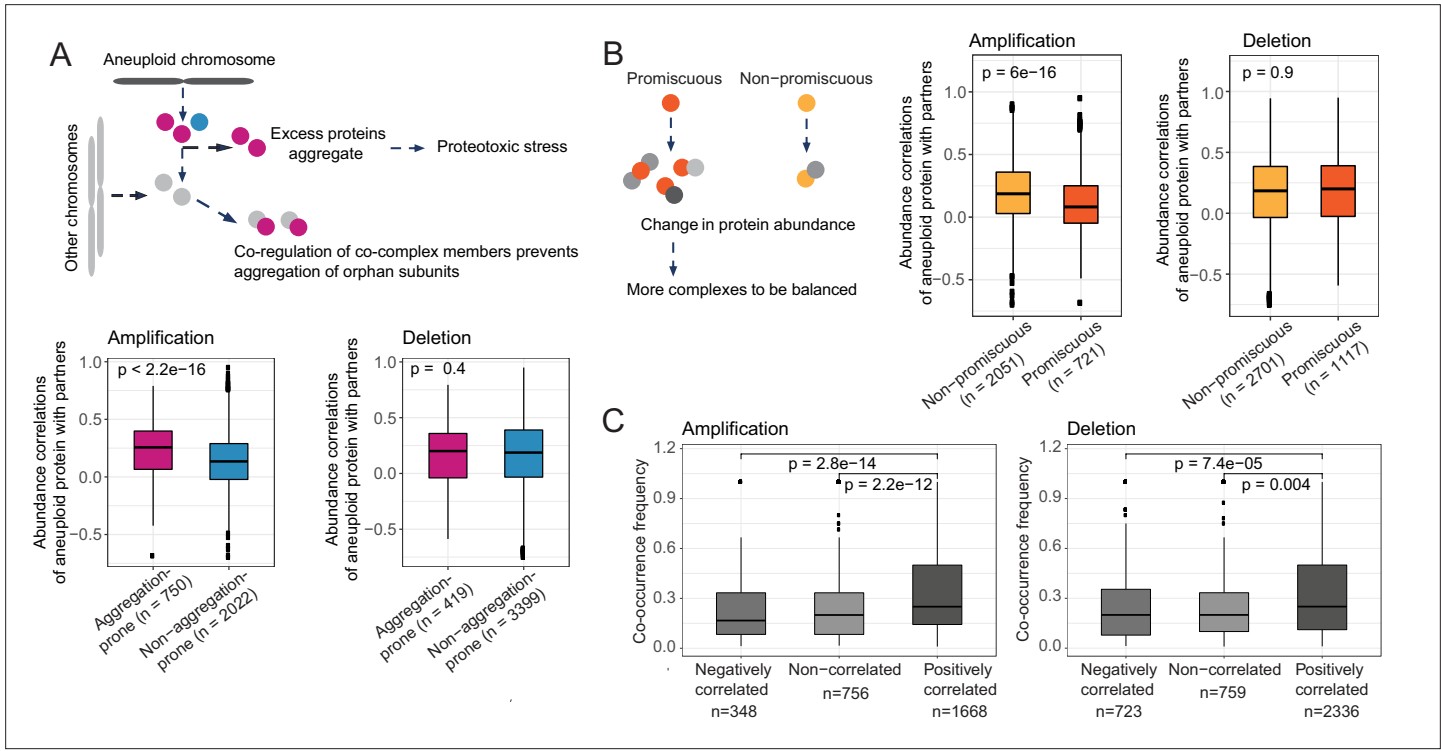

**Figure 4.** Compensatory mechanisms preventing aggregation and imbalances in complexes partly explain abundance changes on other chromosomes. Protein abundance correlations between differentially abundant proteins on aneuploid chromosomes and their co-complex members on other chromosomes when aneuploid proteins are grouped as (**A**) aggregation-prone and non-aggregation-prone and (**B**) promiscuous and non-promiscuous. (**C**) Co-occurrence frequency of differentially abundant proteins on aneuploid chromosomes and their co-complex members on other chromosomes in different correlation groups, positively and negatively correlated and non-correlated co-complex members of aneuploid proteins. Wilcoxon test was used to test the differences between groups.

The online version of this article includes the following figure supplement(s) for figure 4:

**Figure supplement 1.** Aggregation propensity of co-complex members of aneuploid proteins in the case of deletions.

significantly higher numbers of ubiquitination sites (p<0.05, Wilcoxon test; *Figure 3B*). This suggests that a primary mechanism for keeping protein complex stoichiometry in check seems to be indeed post-translational regulation (such as ubiquitin-mediated degradation).

## Differential protein abundance of complex partners as a compensatory mechanism to prevent complex imbalance and aggregation

The main expected detrimental effect of chromosomal amplifications is an excess of protein abundance of complex members leading to an aggregation of the orphan proteins (rather than a loss of function of the complex as would be expected for insufficient expression for complex assembly as a consequence of chromosome deletion) (*Santaguida et al., 2015b*). We therefore tested if aggregation-prone proteins on the amplified chromosome show a higher tendency for strong correlations with their complex partners on other chromosomes. We grouped aneuploid proteins as aggregating and non-aggregating based on the data from *Määttä et al., 2020* and compared their protein-level abundance correlations with partners. Indeed, we observed stronger correlations for aggregation-prone proteins as compared to their non-aggregating counterparts (p<2.2e-16, Wilcoxon test; *Figure 4A*). This suggests that upregulating the protein expression of genes on chromosomes not affected by aneuploidy themselves serves as a compensatory mechanism to prevent proteotoxicity triggered by the aggregation of non-paired complex members located on the aneuploid chromosome.

We hypothesized that in the case of chromosomal deletions, the aggregation propensity of downregulated proteins on the aneuploid chromosome should not affect the degree of correlation with complex partners. Indeed, we observed aggregation-prone proteins to be not related to stronger correlations with their complex partners on other chromosomes when they are encoded on deleted chromosomes (*Figure 4A*). This is likely the case as downregulating those proteins would not leave them as orphan subunits and hence increase their risk of aggregation. However, one would expect that aggregation propensity of co-complex members of downregulated proteins of the deleted chromosomes should have an effect on the co-abundance correlations as this will leave them as orphan subunits. To test this, we compared the co-abundance correlation of proteins of deleted chromosomes with their aggregating co-complex members to non-aggregating ones. We found that deleted aneuploid proteins have significantly stronger correlations with their aggregating co-complex members (p=0.045, Wilcoxon test; *Figure 4—figure supplement 1*).

Assuming that the regulation of proteins on other, non-aneuploid chromosomes serves to prevent stoichiometric imbalance of protein complexes, we speculated that for proteins that are in many complexes there are more ways of being abundance-compensated by a complex partner compared to those proteins participating in few complexes and therefore each single partner should be under less stringent control for co-expression with the aneuploid protein. We therefore classified each aneuploid protein into promiscuous (participating in more than five complexes) and non-promiscuous (involved only in five or less than five complexes). As expected, we observed weaker correlations for promiscuous proteins of amplified chromosomes (p=6e-16, Wilcoxon test; *Figure 4B*) further supporting the model in which differential abundance of proteins on other chromosomes is a compensatory mechanism. We did not observe the same association in the case of chromosomal deletions (*Figure 4B*).

Finally, we hypothesized that proteins co-occurring in many complexes should show stronger correlation than proteins found only in a few cases together in the same complex. Indeed, we found significant differences in the number of times aneuploid proteins and their positively correlated co-complex members were found together in the same complex vs their uncorrelated or negatively correlated co-complex members (p=2.2e-12 and p=2.8e-14 for chromosomal amplifications, p=0.004 and p=7.4e-05 for deletions; *Figure 4C*). This, again, illustrates how complex organization shapes the co-abundance patterns between differentially abundant proteins from the aneuploid chromosome and those located on other chromosomes.

## Functional selection acting on protein stoichiometric imbalance

In the previous sections, we proposed that co-abundance change of protein complex partners is a compensation mechanism to prevent stoichiometric imbalance in protein complexes to avoid proteotoxicity of orphan subunits. We next wondered if besides biophysical (such as aggregation propensity) any functional properties would protect complexes and complex subunits from abundance imbalance in aneuploid cancer cells. To this end, we first retrieved the most strongly correlated co-complex

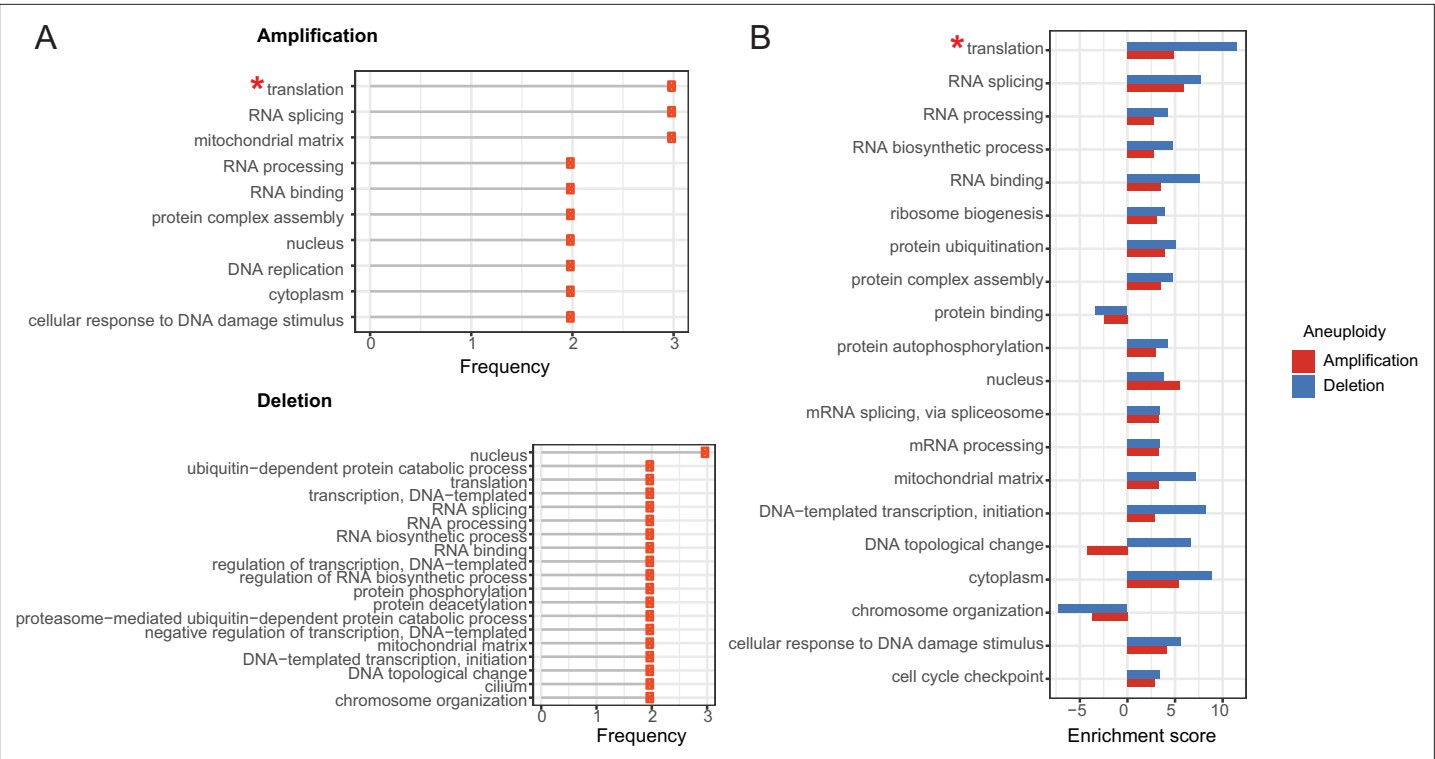

**Figure 5.** Enrichment of functional terms in complexes of top correlated proteins. **A)** Most frequently enriched terms in the amplification and deletion cases. Frequency shows the number of aneuploidy cases in which the corresponding term is significantly enriched (* Enrichment is mostly driven by ribosomal genes). (**B**) Enrichment scores of enriched terms both in amplification and deletion cases. For the functional term that is enriched in more than one amplification/deletion cases, the enrichment score of the ones with the lowest p-value is displayed.

members of differentially abundant aneuploid proteins and identified the complexes they are involved in. Then, we obtained functional annotations of the complexes from the CORUM database. To identify functions under stronger protection from protein abundance imbalance in complexes, we computed the enrichment of these functions compared to a random set of complexes under relaxed stoichiometric protection. The analysis revealed that top correlated proteins form complexes that are frequently involved in translation (mainly driven by ribosomal proteins; see Materials and methods section), RNA splicing, RNA processing, and protein complex assembly (*Figure 5A*; *Supplementary file 6*). Interestingly, the functional enrichment is consistent for amplifications and deletions suggesting that not just compensatory mechanisms to prevent proteotoxicity contribute to the dysregulation of proteins on other chromosomes but also functional selection is in place, acting on important cancer-essential functions up- or downregulating entire protein complexes while keeping their stoichiometry in check.

To quantitatively compare the degree of enrichment between the functional terms associated with balance-protected complexes, we devised an enrichment score (see Materials and methods section) and compared it for the top enriched or depleted functions between amplifications and deletions. We observed that top correlations in the deletion cases are related to stronger enrichment scores when compared to their counterparts in the amplification cases (*Figure 5B*).

## Phenotypic consequences of stoichiometric compensation success

The previous results suggest co-regulation of co-complex members as a compensation mechanism to balance protein abundance changes caused by whole chromosomal alterations and thus to keep protein complex stoichiometry in check. We reasoned that different tumors might be able to compensate for the dysregulation of proteins on the aneuploid chromosome with a different degree of success and hypothesized that tumors that can better compensate for protein abundance changes will be associated with better survival rates while those that fail to compensate should upregulate components of the protein degradation machinery to clear the cell from the orphan complex subunits. To test this, we first calculated a stoichiometry deviation score for each sample by using correlations between

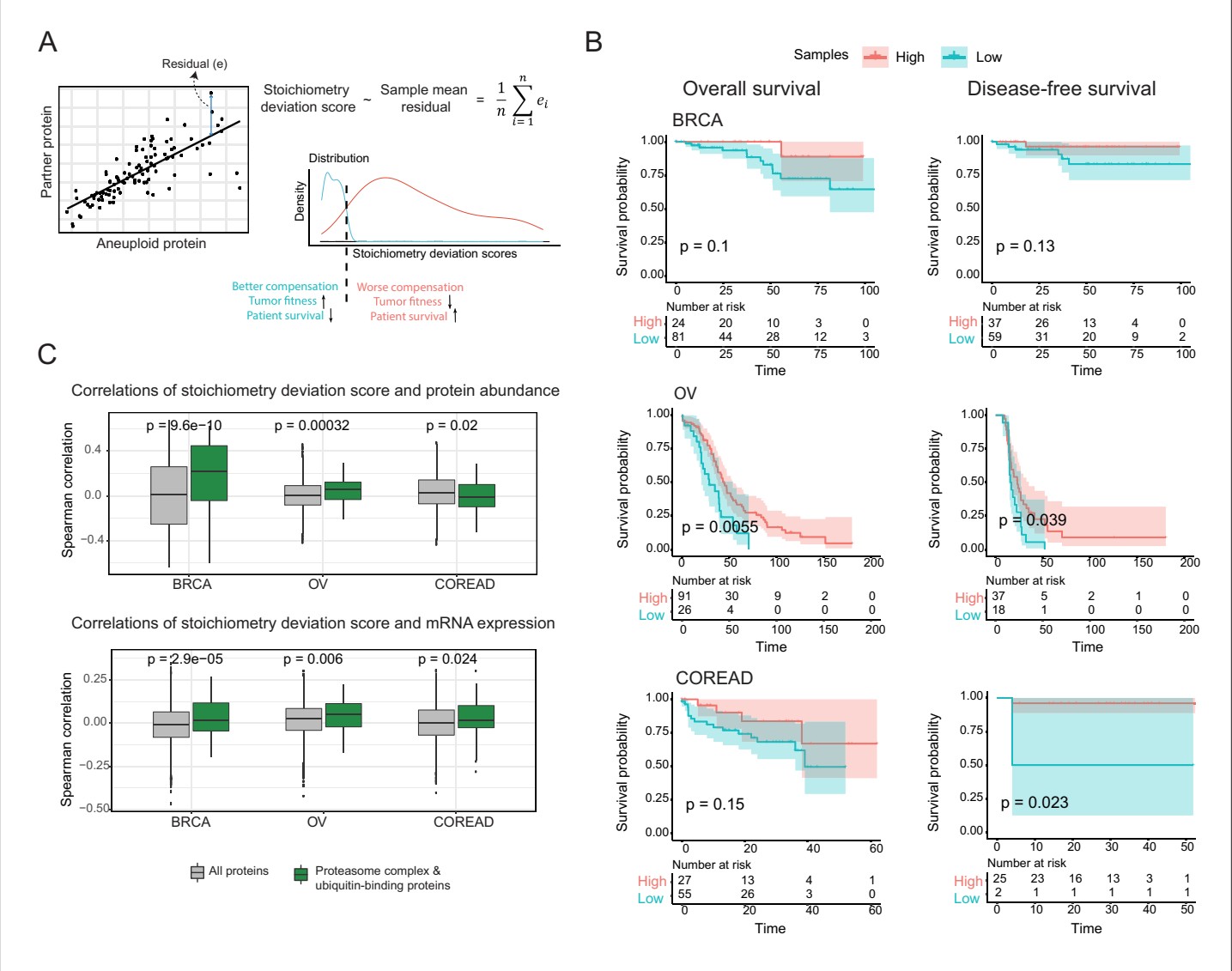

**Figure 6.** Consequences of stoichiometric compensation. (**A**) Graphical representation for the calculation of stoichiometric deviation score for each sample (n=30 referring top 30 correlations). (**B**) Survival analysis results within each tissue. Survival analysis was done once with overall survival and once with disease-free survival. (**C**) Correlations between the stoichiometric deviation scores and protein abundance/mRNA expression of all proteins, and proteasome complex - ubiquitin binding proteins. Wilcoxon test was used to test differences between groups. BRCA, breast cancer; OV, ovarian cancer; COREAD, colorectal adenocarcinoma cancer.

The online version of this article includes the following figure supplement(s) for figure 6:

**Figure supplement 1.** Association between the deviation from complex stoichiometry and survival probability.

co-complex members in aneuploid tumors as a measure of failure of keeping the complex stoichiometry balance (*Figure 6A*). Then, we performed a survival analysis by grouping samples based on their stoichiometry deviation scores (*Figure 6A*). While not significant in every single case we observed a tendency that samples with low stoichiometry deviation scores are related to lower survival probabilities in all three tissue types (*Figure 6B*) showing that compensation for protein abundance indeed provides a survival advantage to tumors.

We further investigated if the proteins that play a role in protein degradation have higher abundances in the tumors that cannot compensate for abundance changes and thus have to deal with the excess amount of orphan subunits. We indeed found that ubiquitin-binding proteins and components of the proteasome show significantly higher correlations between their abundances and the stoichiometry deviation scores in two out of three tissues (*Figure 6C*; *Figure 6—figure supplement*

*1B*), and this tendency still applies when samples are divided into amplification and deletion groups (*Figure 6—figure supplement 1D*). This likely is a consequence of proteotoxic stress resulting from the inability of some tumors to keep protein complexes balanced.

## Discussion

Here, we conduct an extensive characterization of the transcriptome and proteome in aneuploid human cancers. We show that 47–63% and 24–33% of genes on aneuploid chromosomes show expression changes at the transcriptome and proteome level (*Figure 1C*), respectively. A similar degree of the effect of aneuploidy on expression changes has been previously found in human cancer cells: 50% and 25% of genes showed copy-number-correlated expression changes at the transcriptome and proteome level, respectively (*Schukken and Sheltzer, 2021*). In yeasts, aneuploidy affects a larger fraction of genes. Up to 70–80% of genes on aneuploid chromosomes changed levels of transcripts and proteins by the degree expected based on chromosome number (*Dephoure et al., 2014*; *Gasch et al., 2016*). The different degree of expression changes induced by aneuploidy depends on many factors including the cellular environment (*Kojima and Cimini, 2019*). However, independently from the study system, the stronger buffering at the proteome level than transcriptome level is consistently observed (*Stingele et al., 2012*; *Dephoure et al., 2014*). On the other hand, we find that genes on other chromosomes show a surprising degree of differential expression. Further, comparison of transcriptome and proteome data reveals that proteomic changes from aneuploid chromosomes could primarily be explained by differential expression of their corresponding genes at the transcriptome level; however, this is not the case for proteomic changes of other, non-aneuploid chromosomes (in amplification cases, 76.5% and 26.2% of proteomic changes, and in deletion cases, 89.9% and 36.2% of the differentially abundant proteins show also differential abundance of the corresponding transcript for aneuploid and other chromosomes, respectively). Together, these observations suggest different levels of control for dosage compensation for aneuploid and other chromosomes. Transcriptional control plays a major role for aneuploid chromosomes while translational or post-translational control has a comparably stronger importance for gene regulation of other chromosomes.

We propose that a large fraction of differential expression events on other, non-aneuploid chromosomes might serve a compensatory purpose by binding aggregation-prone proteins upregulated due to their location on the aneuploid chromosome. We observe that up to 40% of the typically hundreds and sometimes more than 1000 differentially abundant proteins physically interact either in a complex or in a binary manner with their partners on the aneuploid chromosome. Given the still incomplete understanding of the nature of the human interactome (and in particular the limits on the available protein complex information), this is a remarkably high number. We expect that with an increase of protein complex measurements, this number will substantially grow.

This novel compensatory mechanism complements the previously described dosage compensation addressing the differential expression of complex subunits directly on the aneuploid chromosomes (*Stingele et al., 2012*; *Schukken and Sheltzer, 2021*). Together they might largely prevent the otherwise detrimental overexpression of orphan complex subunits. Correlated abundance patterns to compensate for aggregation-prone orphan proteins (differentially abundant aggregation-prone proteins of amplified chromosomes and aggregation-prone co-complex members of differentially abundant proteins of deleted chromosomes, respectively, for amplification and deletion cases) are detectable for both chromosome amplification and deletion cases. Furthermore, we observed stronger enrichments for specific, cancer-essential functions suggesting that here functional selection is a stronger driving force to shape the global co-abundance pattern. Among these terms, enrichment of translation is mostly driven by a relatively larger fraction of ribosomal genes among our gene sets (which might not be surprising given that 23% of the translation-related genes are ribosomal genes).

Similar compensatory mechanisms have been previously identified to be induced by focal CNAs (*Gonçalves et al., 2017*; *Sousa et al., 2019*). We demonstrate here that this observation holds for large genomic amplifications of entire chromosomes and likely serves the prevention of proteotoxic aggregation of orphan subunits as suggested by the stronger abundance correlations formed by aggregation-prone proteins. Considering that around 90% of solid tumors are aneuploid, our work addresses the question of how the vast gene expression changes induced by the amplification of large genomic regions can be tolerated by the majority of cancer cells.

One surprising observation is the presence of strong negative correlations between aneuploid proteins and co-complex partners on other chromosomes. We could not substantiate our initial intuition that those could be indicative of protein-binding competition relationships. We used different approaches to predict overlap in protein-binding interfaces to estimate competition events but did not observe an agreement with the negative abundance correlations. Future research will need to clarify the reason for the existence of the negative correlations.

Our findings describe the need for compensation mechanisms to deal with stoichiometric imbalances in protein complexes induced by aneuploidy and highlight the role of components of proteasome complex and ubiquitin-binding proteins in keeping the complex stoichiometry and better tumor fitness. It has been shown that targeting essential genes in aneuploid cells eventually results in proliferation defects and activation of cell death pathways (*Cohen-Sharir et al., 2021*). Taken together, our results could serve for the identification potential drug targets for clinical use. Ultimately, given the high number of aneuploid tumors, studying and understanding compensatory mechanisms and the potential vulnerabilities they create in aneuploid tumors will have profound implications for both basic cell biology as well as cancer biology.

## Materials and methods

### Calculating whole-chromosome-level aneuploidy scores

Arm-level aneuploidy scores for 10,522 TCGA samples, comprising 33 cancer types, were obtained from *Taylor et al., 2018*. Whole-chromosome-level aneuploidy scores were calculated as follows: If both p and q arms for chromosomes 1–12 and 16–20 are amplified, deleted, or not changed, the entire chromosome was considered as amplified, deleted, or diploid, respectively. For acrocentric chromosomes, 13–15 and 21–22, q arm aneuploidy scores were considered as representative for whole-chromosome-level aneuploidy scores (*Supplementary file 1*). TCGA samples that have conflicting events (amplification, deletion, or no change) on different arms or missing data for one or both arms were removed from further analyses. In this study, colon (COAD) and rectum adenocarcinoma (READ) were considered as one cancer type as COREAD.

### Detecting cancer-type-specific whole-chromosome-level aneuploidies

Chi-square test was performed to test the occurrence of a whole-chromosome-level aneuploidy within each cancer type against random expectation. Then multiple testing correction was applied on the p-values by using Holm's method. Cancer-type-specific, whole-chromosome-level aneuploidies were selected based on the criteria that adjusted p-value lower than or equal to 0.05 and chi-square standard residual equal to or higher than 2, resulting in 86 and 117 whole-chromosome-level amplifications and deletions, respectively (*Supplementary file 1*).

For each of the 86 cancer-type-specific amplifications, co-amplification frequency with other chromosomes was tested by using chi-square test. Then multiple testing correction was applied on the p-values by using Holm's method. 305 significant combinations were identified as co-amplified events in 60 out of the 86 cancer-type-specific amplifications based on the criteria that the adjusted p-value was lower than 0.01 (*Supplementary file 1*). Then the contribution of each chromosome to the transcriptional dysregulation on other, non-aneuploid chromosomes was calculated as the percentage of differentially expressed genes by dividing the number of differentially expressed genes to the total number of genes on that chromosome. A paired Wilcoxon test was used to compare mean contribution of co-amplified chromosomes to that of non-co-amplified chromosomes across 60 cancer-type-specific events.

### Data processing

RNA-seq fragments per kilobase of exon per million reads mapped (FPKM) values for 11,007 TCGA samples, comprising 32 cancer types, were downloaded from the NCI Genomic Data Commons (GDC) (*Grossman et al., 2016*). Then FPKM values were converted to transcripts per million (TPM) values, and primary tumor samples (n=9830) were selected. Ensembl gene IDs were mapped to gene symbols based on the mapping obtained from ensembl BioMart (Human genome version GRCh38.p13 - downloaded on May, 2019) (*Howe et al., 2021*), and the mean value was taken when multiple Ensembl

IDs mapped to one gene symbol. Mitochondrial and non-expressed (zero values in all samples) genes were removed.

Proteomics data used in this publication were generated by the CPTAC (NCI/NIH). Proteomics measurements for the available TCGA projects were downloaded from the CPTAC, covering three cancer types, spectral counts for COREAD (*The Cancer Genome Atlas Network, 2012*; *the NCI CPTAC et al., 2014*) (90 samples and 5561 genes), and relative abundances for OV (*Cancer Genome Atlas Research Network, 2011*; *Zhang et al., 2016*) (174 samples and 7169 genes), and BRCA (*Cancer Genome Atlas Network, 2012*; *Mertins et al., 2016*) (105 samples and 10,625 genes). For the replicated samples, the mean value was considered. Primary samples covered by the transcriptomic data and genes that are expressed at transcriptome level (genes having TPM value in at least one sample) were selected, which gave us 88 samples and 5353 genes, 119 samples and 7062 genes, 105 samples, and 10,467 genes for COREAD, OV, and BRCA, respectively (*Figure 1—figure supplement 1A*). Spectral counts for COREAD were normalized by quantile normalization followed by log2 transformation.

## Detecting transcriptomic and proteomic changes

To detect transcriptomic and proteomic changes, samples covered by aneuploidy, transcriptomic, and proteomic data were selected, which resulted in 9266 samples for transcriptome analysis and 298 samples for proteome analysis.

For each of the 203 cancer-type-specific, whole-chromosome-level aneuploidies covering 86 amplifications and 117 deletions, we first grouped TCGA samples as the ones with chromosome amplification/deletion and the ones diploid for the respective chromosome. After selecting the samples, genes having zero TPM in all samples were filtered out. Differentially expressed genes between the samples with diploid and those with an altered chromosome were identified by using Wilcoxon test (we consistently used Wilcoxon test to identify transcriptomic and proteomic changes to avoid detection biases introduced by applying different methods to different data types) and then multiple testing correction was performed on the p-values by using the Benjamini and Hochberg method. Significantly, differentially expressed genes were selected based on the criteria that the adjusted p-value is lower than 0.1. For the cases where we were left with less than 250 differentially expressed protein coding genes after adjusted p-value cutoff, the uncorrected p-value was used (p<0.05) in order to have a sufficient number of genes to perform the enrichment tests (described below).

For the 13 cancer-type-specific, whole-chromosome-level amplifications and 20 cancer-type-specific, whole-chromosome-level deletions covering COREAD, OV, and BRCA cancer types, for which the corresponding proteome data is available, differentially abundant proteins between the samples with diploid- and amplified/deleted chromosome were detected by using Wilcoxon test. Proteins with a p-value lower than 0.1 were considered as significantly differentially abundant (again, using a relaxed statistical cutoff in order to perform the subsequent analyses of the protein set).

To dissect frequently dysregulated genes in aneuploidy and their associated molecular functions, we, first, counted how many times a gene was dysregulated across different aneuploidy cases (203 and 33 detected cancer-type-specific, whole-chromosome-level aneuploidies, respectively, for transcriptomic and proteomic data). Then we performed GO analysis on the most frequently dysregulated 150 genes in amplification and deletion cases by using WebGestalt (*Liao et al., 2019*). The GO analysis was performed separately for the gene sets from transcriptomic and proteomic data. All protein coding genes were used as a background.

## Grouping proteins and protein pairs

Aggregation-prone proteins (n=300) were obtained from *Määttä et al., 2020*. The known human protein complexes (n=2916) were downloaded from the CORUM database (*Giurgiu et al., 2019*) (CORUM 3.0; September, 2018). The number of complexes a protein is involved in was calculated by considering the CORUM complexes, and then proteins were grouped as promiscuous if they are involved in more than five complexes, otherwise as non-promiscuous. To calculate co-occurrence frequencies, for each protein pair, we first counted the number of complexes in which the two proteins were found together and then divided it by the number of complexes in which at least one of them is found.

## Statistical analyses

Chi-square test was used to assess the relationship between differentially abundant proteins on other chromosomes and (i) complex members obtained from the CORUM database, and (ii) co-complex members of differentially abundant proteins on aneuploid chromosomes for the 13 cancer-type-specific amplifications and 20 deletions. The same association tests were repeated by using transcriptome-level changes - differentially expressed genes on other chromosomes - for 86 amplifications and 117 deletions, and then multiple testing correction was performed (only for transcriptome-level analysis as here a much larger number of tests was performed) on p-values by using Holm's method.

Cancer-type-specific protein abundance correlations were calculated between all possible pairs across primary tumor samples for the cancer types BRCA, COREAD, and OV (for which we have proteomic measurements) by using the Spearman method. Comparing correlations between co-complex members and non-complex members was done by considering differentially abundant proteins on aneuploid chromosomes and their correlations with all CORUM complex subunits. The Wilcoxon test was used to compare the correlation distributions. To this end, correlations from three cancer types were pooled.

To compare the protein-level correlations between different protein feature groups, correlations between differentially abundant proteins on aneuploid chromosomes (the ones that showed an increase in abundance for amplifications or a decrease for deletions) and their co-complex members were considered. To obtain a unique set, correlations from three cancer types were pooled. For the pairs for which we could compute a correlation in more than one cancer type, the maximum correlation value was considered, which left us with 2772 and 3818 correlations for amplifications and deletions, respectively (*Supplementary file 5*).

## Network randomization

To assess if there is an enrichment between differentially abundant proteins encoded on the aneuploid and those on other chromosomes, we employed a randomization test. We retrieved PPI data from HIPPIE (v2.2) (*Alanis-Lobato et al., 2017*) and counted the number of physical PPIs between the two protein sets. In each randomization, we replaced the set of aneuploid differentially abundant proteins by a protein set of equal size. To avoid biases, we additionally enforced the same degree distribution as in the original set by replacing each differentially abundant aneuploid protein by a protein of the same or similar degree (forming as many interactions as the replaced protein). We then recounted the number of PPIs between the random set and differentially abundant proteins encoded on other chromosomes to construct a background distribution from which we estimated the p-value by counting how often the original observed value was smaller than or equal to a randomized value (*Supplementary file 4*).

## Functional annotation of protein complexes

To investigate functional relevance of co-abundance regulation, we first classified abundance correlations between differentially abundant proteins of aneuploid chromosomes and their co-complex members of other chromosomes into two groups: Top correlated ones including 20 strongest positive and negative correlations (40 in total) and non-correlated ones including correlations between –0.2 and 0.2. The latter was used as background in the association test. Then we obtained protein complexes and their functional annotations - associated GO terms - from the CORUM human protein complex data. For each GO term, a chi-square test was performed in which the number of complexes related to the corresponding term in the top correlated group was tested against that of in the background group. An enrichment score was calculated by dividing the difference of the observed complex number and the expected one obtained from the chi-square test by the square root of the expected value. GO terms with p-value lower than 0.05 were considered as significantly associated. The analysis was done separately for each detected cancer-type-specific aneuploidy case (13 amplifications and 20 deletions covering BRCA, OV, and COREAD cancer types for which we have proteomic measurements).

As we observed an enrichment of post-transcriptional- and translation-related functions, which often involve larger complexes, we further aimed to test how much this enrichment is driven by larger complexes and a large number of ribosomal genes. To end this, we repeated the test for functional enrichment once after removing ribosomal genes downloaded from the HGNC database

(*Tweedie et al., 2021*), and once after removing large complexes (complexes that include more than 10 subunits). This did not change the results much indicating that the observed enrichment did not depend on large complexes only.

## DNA methylation analysis

Promoter-level methylation measurements calculated from probe-level methylation data in TCGA were used in this analysis (*Heery and Schaefer, 2021*). For each gene, the most upstream promoter was considered for the analysis. Average methylation level of genes was calculated by taking the mean of methylation levels across aneuploid samples and diploid samples, separately. Wilcoxon tests were performed for the statistical comparison between up- and downregulated genes.

## TF - target randomization test

To test if differential expression of TFs on aneuploid chromosomes could explain the vast expression changes on other chromosomes, we performed a randomization test for 203 detected cancer-type-specific aneuploidies. We first retrieved 1651393 ENCODE gene-TF associations detected by ChIP-Seq experiments from the Harmonizome database (*ENCODE Project Consortium, 2004*; *Rouillard et al., 2016*) and then counted the number of targets of differentially expressed TFs on aneuploid chromosomes among differentially expressed genes on other chromosomes. To compute a background distribution, we recounted the number of targets for random TF sets for 100 iterations. In each iteration, the random TF set size was equal to the number of differentially expressed TFs on the aneuploid chromosome of the corresponding cancer-type-specific aneuploidy case. The p-value was calculated using the background distribution by conducting a two-tailed test.

To quantify the degree of TF regulation on the transcript changes of co-complex members in aneuploid tumors, we counted the number of targets among differentially expressed co-complex members of aneuploid proteins on other chromosomes. The aneuploidy cases in BRCA, COREAD, and OV (where we have both transcriptomic and proteomic data) were considered for the analysis to make the results comparable. To test the degree of TF regulation on overall dysregulation in tumor vs normal, targets were counted among all dysregulated genes in tumor vs normal.

## Ubiquitination analysis

Experimentally observed ubiquitination sites for human proteins were downloaded from PhosphoSitePlus (*Hornbeck et al., 2015*). The unique set of abundance correlations between differentially abundant proteins of aneuploid chromosomes and their co-complex members of other chromosomes was used for this analysis (See Materials and methods section: Statistical analyses; *Supplementary file 5*). For each co-complex member protein, the total number of ubiquitination sites was calculated as the sum of all ubiquitination sites. The proteins not covered by the ubiquitination dataset were removed from the analysis. Abundance correlations equal to or higher than 0.4 and lower than or equal to –0.4 were considered as top positive and top negative correlations, respectively. Wilcoxon test was used for statistical comparison.

## Calculation of the stoichiometry deviation score and survival analysis

To calculate the stoichiometry deviation score for each TCGA sample, the top 30 strongest tissue-specific correlation pairs (aneuploid protein and its partner) in amplification cases were taken. For each pair, a linear regression model was performed in which protein abundance of partner protein was dependent variable, and that of aneuploid protein was independent variable. Then the stoichiometry deviation score of a sample was calculated as the mean of absolute residuals in the regression models.

To perform survival analysis, we first grouped samples into two sets (high and low) based on their stoichiometry deviation scores by using the survminer package (version 0.4.8) in R, which is using the maximally selected rank statistics to determine the optimal cutpoint. Samples with the deviation scores lower than or equal to the cutpoint were assigned as the low group; otherwise they were assigned to the high group (*Figure 6—figure supplement 1C*). Then, we performed survival analysis once with overall survival and once with disease-free survival by using Kaplan Meier method in the survival package (version 3.1.8) in R.

GO annotations were retrieved from the UniProt database, and proteins with GO terms related to proteasome complex and ubiquitin-binding were selected. Then, the correlation between protein

abundances and stoichiometry deviation scores was calculated once across all samples and once for amplification and deletion groups, separately by using the Spearman method. Samples were separated into amplification and deletion groups if they are with at least one detected cancer-type-specific chromosomal amplification and deletion, respectively. Then, the Wilcoxon test was used for the comparison.

## Availability of data and materials

The code performing all analyses in this study is available at https://github.com/SengerG/Coregulation-of-complexes-in-Aneuploidtumors.git, (copy archived at swh:1:rev:b9ef349854c073044c2ecb-f743819e12abb17923, *Senger, 2022*).

## Acknowledgements

We would like to thank Jason M Sheltzer for his careful reading of our manuscript and very helpful comments. Gökçe Senger is a PhD student within the European School of Molecular Medicine (SEMM). The work leading to this manuscript was supported by Fondazione AIRC, grant reference number MFAG 21791 and partially supported by the Italian Ministry of Health with Ricerca Corrente and 5x1000 funds. Work in the Santaguida lab is supported by grants from the Italian Association for Cancer Research (MFAG 2018 - ID. 21,665 project), Ricerca Finalizzata (GR-2018–12367077), Fondazione Cariplo, the Rita-Levi Montalcini program from MIUR, and the Italian Ministry of Health with Ricerca Corrente and 5×1000 funds.

## Additional information

### Funding

| Funder | Grant reference number | Author |
| --- | --- | --- |
| Fondazione AIRC | MFAG 21791 | Martin H Schaefer |

The funders had no role in study design, data collection and interpretation, or the decision to submit the work for publication.

### Author contributions

Gökçe Senger, Conceptualization, Data curation, Formal analysis, Investigation, Methodology, Software, Visualization, Writing - original draft, Writing - review and editing; Stefano Santaguida, Conceptualization, Writing - review and editing; Martin H Schaefer, Conceptualization, Funding acquisition, Investigation, Methodology, Project administration, Resources, Supervision, Writing - original draft, Writing - review and editing

### Author ORCIDs

Stefano Santaguida http://orcid.org/0000-0002-1501-6190
Martin H Schaefer http://orcid.org/0000-0001-7503-6364

### Decision letter and Author response

Decision letter https://doi.org/10.7554/eLife.75526.sa1
Author response https://doi.org/10.7554/eLife.75526.sa2

## Additional files

### Supplementary files

• Supplementary file 1. Whole-chromosome-level aneuploidy scores, cancer-type-specific whole-chromosome-level amplifications and deletions, and co-amplified chromosomes, related to *Figure 1* and *Figure 1—figure supplement 1B*.

• Supplementary file 2. Frequently dysregulated genes on other chromosomes and their associated GO terms.

• Supplementary file 3. The number of differentially abundant proteins on other chromosomes

in partners of differentially abundant aneuploid chromosomes, in all expressed proteins, and in CORUM subunits. Standard residuals and p-values for the overlap between differentially abundant proteins on other chromosomes and partners of differentially abundant aneuploid proteins.

• Supplementary file 4. Network randomization results; the number of PPIs between differentially abundant proteins on aneuploid chromosomes and those on other chromosomes and their corresponding p-values, related to *Figure 2C*.

• Supplementary file 5. Protein-level Spearman correlations between differentially abundant proteins on amplified/deleted chromosomes and their co-complex members on other chromosomes, and groups of proteins and protein pairs, related to *Figure 4*.

• Supplementary file 6. Functional enrichment analysis of protein complexes; significantly enriched GO terms in different aneuploidy cases, their related p-values and enrichment scores, related to *Figure 5*.

• Transparent reporting form

### Data availability

The code performing all analyses in this study is available at https://github.com/SengerG/Coregulation-of-complexes-in-Aneuploidtumors.git, (copy archived at swh:1:rev:b9ef349854c073044c2ecbf743819e12abb17923).

The following dataset was generated:

| Author(s) | Year | Dataset title | Dataset URL | Database and Identifier |
|---|---|---|---|---|
| Senger G | 2021 | Coregulation-ofcomplexes-inaneuploidtumors | https://github.com/SengerG/Coregulation-of-complexes-in-Aneuploidtumors | GitHub, b9ef349 |

The following previously published dataset was used:

| Author(s) | Year | Dataset title | Dataset URL | Database and Identifier |
|---|---|---|---|---|
| Rouillard AD, Gundersen GW, Fernandez NF, Monteiro CD, McDermott MG, Ma'ayan A | 2016 | ENCODE Transcription factor targets | https://maayanlab.cloud/Harmonizome/dataset/ENCODE+Transcription+Factor+Targets | Harmonizome, Harmonizome |

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
