## [Editor Report]

This paper will be of interest to the cancer biology community. The study leverages high-throughput genomic and proteomic data to evaluate the role of aneuploidy on functional pathway changes in cancer.

---

## [Decision Letter]

**Decision letter after peer review:**

Thank you for submitting your article "Regulation of protein complex partners as a compensatory mechanism in aneuploid tumors" for consideration by *eLife*. Your article has been reviewed by 3 peer reviewers, including Elana J Fertig as Reviewing Editor and Reviewer #1, and the evaluation has been overseen by Naama Barkai as the Senior Editor. The following individuals involved in review of your submission have agreed to reveal their identity: Erik McShane (Reviewer #2); Elsa Logarinho (Reviewer #3).

Essential revisions:

1) Dissect for specific genes / proteins in the differential expression / abundance in Figure 1.

2) Define clearly the threshold used in survival analysis.

3) Consider more standard genomics analysis tools in the analysis.

4) Enhance analysis methods, including as examples the suggestions below:

– Exclude ribosomal subunits in the functional analysis.

– Look at the transcript levels to assess the degree of impact on the expression changes of complex partners in figure 5.

– Overlap data with previous data on post-transcriptional regulation.

– Better crosstalk the transcriptome and proteome data.

– Investigate if the complex partners on other chromosomes, in the case of deletions, are aggregation-prone.

– Check if correlations in figure 6 still apply if the tumors' samples are separated in amplification and deletion groups.

*Reviewer #1 (Recommendations for the authors):*

Figure 1 focuses on the number of differential expression / differential abundance changes but not the molecular pathways associated with aneuploidy. Some discussion of the specific genes / proteins that were found and their biology would be helpful.

It would be helpful to contextualize the enrichment of transcription factors and methylation relative to their association globally independent of the context of aneuploidy to validate some of the negative claims of the manuscript.

There are concerns about the survival analysis (e.g., how censoring was accounted for in the high vs low survival groups), and the clarity of the definition for thresholding used to separate sample groups in the Kaplan Meir analysis.

*Reviewer #2 (Recommendations for the authors):*

Overall, I think this manuscript is really interesting and I think it is a great addition to the field. I have some specific questions and suggestions that I hope will clarify and improve the, in my opinion, few weaknesses of this manuscript.

1) Could the test on the effect of co-amplification be done in a more systematic manner instead of just looking at chromosome 7 amplification in thyroid cancer?

a) Perhaps just by looking at more specific cases?

2) The functional analysis leaves many questions that should be discussed:

a) How much of the functional analysis results is driven by the cyto and mitochondrial ribosomes? Since they are very large and abundant complexes, which have previously been shown to be post-translationally regulated (Bogenhagen and Haley 2020; Lam et al., 2007), they have a tendency to overwhelm analyses dependent on proteomics data. It would be nice to see the same analysis when excluding ribosomal subunits.

b) In addition, I am not certain that a functional selection is needed to have a functional enrichment that is consistent between amplifications and deletions. Many of the terms involve large complexes that include many aggregation-prone subunits. Because the abundance of many of these subunits are commonly regulated by degradation also in healthy tissues it is not surprising to see these terms pop up. One could maybe try to separate a functional selection from indirect effects of these functional terms already being enriched for proteins that are prone to be post-translationally regulated (e.g. non-exponentially degraded proteins in (McShane et al., 2016)) by overlapping your data with the list of subunits in the aforementioned paper?

3) I think that the methylation analysis (paragraph starting on line 255) is good but would be better served if it came at an earlier point in the manuscript as it does not directly look at protein complex subunits. E.g. "we could not explain all the variability by methylation patterns therefore we looked at post translational mechanisms [follow protein complex subunit analysis. Figure 2]".

a) The same goes for the TF analysis as it does not directly look at or protein complex subunits.

b) Rather I think the authors could directly look at the transcript levels to assess the degree of impact of transcriptional regulation on the expression changes of complex partners? This would be a powerful analysis better suited for figure 5.

4) Panel 5B protein ubiquitination is quite nice, but I have not seen any correlation between the number of ubiquitination sites and the propensity to degrade fast. The authors could perhaps reference some literature for this or do their own correlation analysis between turnover rates and number of ubiquitination-sites.

a) In addition, it would be better to leave out proteins lacking ubiquitination-sites in the database rather than assume they're lacking (and thus give them a 0 value) as this would bias the data heavily towards very abundant proteins, such as the ribosome.

b) There is some good literature on the E3 ubiquitin ligase (Yanagitani, Juszkiewicz, and Hegde 2017; Nguyen et al., 2017) that targets orphan subunits and ubiquitinates proteins at sites that are usually protected while they're in a complex. Looking at ubiquitination sites in interaction surfaces would be a more mechanistically attractive analysis but probably less feasible?

References:

Bogenhagen, Daniel F., and John D. Haley. 2020. "Pulse-Chase SILAC-Based Analyses Reveal Selective over-Synthesis and Rapid Turnover of Mitochondrial Protein Components of Respiratory Complexes." The Journal of Biological Chemistry, January. https://doi.org/10.1074/jbc.RA119.011791.

Dephoure, Noah, Sunyoung Hwang, Ciara O'Sullivan, Stacie E. Dodgson, Steven P. Gygi, Angelika Amon, and Eduardo M. Torres. 2014. "Quantitative Proteomic Analysis Reveals Posttranslational Responses to Aneuploidy in Yeast." *eLife* 3 (July): e03023.

Lam, Yun Wah, Angus I. Lamond, Matthias Mann, and Jens S. Andersen. 2007. "Analysis of Nucleolar Protein Dynamics Reveals the Nuclear Degradation of Ribosomal Proteins." Current Biology: CB 17 (9): 749-60.

McShane, Erik, Celine Sin, Henrik Zauber, Jonathan N. Wells, Neysan Donnelly, Xi Wang, Jingyi Hou, et al., 2016. "Kinetic Analysis of Protein Stability Reveals Age-Dependent Degradation." Cell 167 (3): 803-15.e21.

Nguyen, Anthony T., Miguel A. Prado, Paul J. Schmidt, Anoop K. Sendamarai, Joshua T. Wilson-Grady, Mingwei Min, Dean R. Campagna, et al., 2017. “UBE2O Remodels the Proteome during Terminal Erythroid Differentiation.” Science 357 (6350). https://doi.org/10.1126/science.aan0218.

Yanagitani, Kota, Szymon Juszkiewicz, and Ramanujan S. Hegde. 2017. “UBE2O Is a Quality Control Factor for Orphans of Multiprotein Complexes.” Science 357 (6350): 472-75.

---

## [Author Response]

Reviewer #1 (Recommendations for the authors):Figure 1 focuses on the number of differential expression / differential abundance changes but not the molecular pathways associated with aneuploidy. Some discussion of the specific genes / proteins that were found and their biology would be helpful.

To identify specific genes that are frequently dysregulated in aneuploid tumors, and their associated molecular functions, we performed GO analysis on the most frequently changed genes and proteins, respectively in transcriptomic and proteomic data. We extended the Results section (page 5 – line numbers 4-8) including the discussion on frequently dysregulated molecular functions in aneuploid tumors and the Materials and methods section (page 14 – line numbers 35-37 and page 15 – line numbers 1-4). We provided the list of top dysregulated genes, proteins, and the results of the GO analysis in Supplementary file 2.

It would be helpful to contextualize the enrichment of transcription factors and methylation relative to their association globally independent of the context of aneuploidy to validate some of the negative claims of the manuscript.

To test if in general our measures of transcriptional regulation are associated with differential expression in cancer (independently from aneuploidy), we first detected differentially expressed genes between tumor and normal samples (in 21 cancer types). Then,

1. We tested if down- and upregulated genes show different methylation levels in tumors when compared to normal. We observed that downregulated genes are mostly related to higher methylation levels in tumor samples, while there is a decrease in the methylation level of upregulated genes in tumor samples.

2. We tested if the targets of differentially expressed transcription factors are enriched among differential expressed genes between tumor vs. normal. Overall, we observed an increased tendency towards enrichment of the targets among dysregulated genes in tumor samples but these were not significant.

We now added a supplementary figure (Figure 3—figure supplement 2A, B) and incorporated these analyses in the corresponding Results section (page 7 – line numbers 23-38).

There are concerns about the survival analysis (e.g., how censoring was accounted for in the high vs low survival groups), and the clarity of the definition for thresholding used to separate sample groups in the Kaplan Meir analysis.

In the survival analysis, the R package “survminer”, which we used to group samples as high and low, uses the maximally selected rank statistics to find the optimal cutpoint giving the most significant relationship between the groups. We clarify this now in the Materials and methods (page 18 – line numbers 7-12) section and show cutpoints for each cancer type in Figure 6—figure supplement 1C.

Reviewer #2 (Recommendations for the authors):Overall, I think this manuscript is really interesting and I think it is a great addition to the field. I have some specific questions and suggestions that I hope will clarify and improve the, in my opinion, few weaknesses of this manuscript.1) Could the test on the effect of co-amplification be done in a more systematic manner instead of just looking at chromosome 7 amplification in thyroid cancer?a) Perhaps just by looking at more specific cases?

We identified 305 co-amplification events for 60 cancer-type-specific amplifications (e.g. chr 2 and chr 20 co-amplification in STAD chr7 amplified tumors). To systematically test if these co-amplification events could explain the expression changes on other chromosomes, we, first, grouped chromosomes as co-amplified and non-co-amplified for each cancer-type-specific amplification. Then we compared the mean percentage of differentially expressed genes on co-amplified chromosomes to that of non-co-amplified chromosomes across 60 cancer-type-specific aneuploidies. We found that there is no significant difference between the medians of these two groups (paired Wilcoxon test, p-value = 0.4) suggesting that the co-amplified chromosomes do not contribute more strongly to the overall transcriptional changes in aneuploid cells as compared to non-co-amplified chromosomes. The corresponding changes in the Results (page 4 – line numbers 20-30) and Materials and methods (page 13 – line numbers 21-27) sections were done. In addition, Figure 1—figure supplement 1 was revised in the direction of this comment and suggestions from reviewer 3, and the corresponding changes were made in the figure legend (page 20 – Figure 1—figure supplement 1 legend).

2) The functional analysis leaves many questions that should be discussed:a) How much of the functional analysis results is driven by the cyto and mitochondrial ribosomes? Since they are very large and abundant complexes, which have previously been shown to be post-translationally regulated (Bogenhagen and Haley 2020; Lam et al., 2007), they have a tendency to overwhelm analyses dependent on proteomics data. It would be nice to see the same analysis when excluding ribosomal subunits.

To further understand if the relatively higher number of ribosomal genes biased the functional analysis, we repeated the analysis after removing ribosomal genes. Overall, the results (see Author response figure 1) were consistent with the previous results indicating that the previously observed functional enrichment did not depend strongly on ribosomal genes. One difference was however that after removing ribosomal genes, the term translation was not significant anymore. This might not be surprising as 23% of all translation-annotated genes are in fact ribosomal genes. We mention now in the manuscript that the functional enrichment of the term “translation” depends on the ribosomal genes (and highlight the term in Figure 5 – previous Figure 4) and hence, as the reviewer points out, might be a consequence of them dominating proteomics analyses (page 10 – line number 7; page 12 – line numbers 9-12; page 19 – Figure 5 legend). In addition, we extended the Materials and methods section explaining our methodology to remove ribosomal genes (page 16 – line numbers 29-35).

**Author response image 1. sa2fig1:** The enrichment of functional terms in complexes of top correlated partners of aneuploid proteins after removing ribosomal genes. Frequency shows the number of aneuploidy cases in which the corresponding term is enriched.

b) In addition, I am not certain that a functional selection is needed to have a functional enrichment that is consistent between amplifications and deletions. Many of the terms involve large complexes that include many aggregation-prone subunits. Because the abundance of many of these subunits are commonly regulated by degradation also in healthy tissues it is not surprising to see these terms pop up. One could maybe try to separate a functional selection from indirect effects of these functional terms already being enriched for proteins that are prone to be post-translationally regulated (e.g. non-exponentially degraded proteins in (McShane et al., 2016)) by overlapping your data with the list of subunits in the aforementioned paper?

To assess at which degree this affected our functional annotation analysis, we, first, tested the overlap between the proteins in our data and non-exponentially degraded (NED) proteins defined by McShane and his colleagues, as suggested. We found that the overall percentage of NED proteins in our data is 10% which is not higher than the pre-defined ratio of NEDs among all proteins by McShane et al.,3. Furthermore, the ratio is mainly driven by the aneuploidy cases where the overlap with ribosomal proteins is high as many of them are also classified as NEDs (The effect of removing ribosomal proteins was addressed in the reply to the previous comment). To further investigate the effect of NEDs in our functional analysis, we removed those proteins from our gene sets, and repeated the analysis. This did not change the results except that the enrichment of the term translation disappeared (see Author response figure 2).

We, then, repeated the functional analysis by removing very large complexes (those with more than 10 subunits). This did not change the enrichment of post-transcriptional- and (post)translational-related terms in which terms such as RNA processing, regulation of translation, protein complex assembly, and RNA splicing still come up.

**Author response image 2. sa2fig2:** The enrichment of functional terms in complexes of top correlated partners of aneuploid proteins after removing NEDs. Frequency shows the number of aneuploidy cases in which the corresponding term is enriched.

3) I think that the methylation analysis (paragraph starting on line 255) is good but would be better served if it came at an earlier point in the manuscript as it does not directly look at protein complex subunits. E.g. "we could not explain all the variability by methylation patterns therefore we looked at post translational mechanisms [follow protein complex subunit analysis. Figure 2]".a) The same goes for the TF analysis as it does not directly look at or protein complex subunits.

We agree that moving the methylation and TF analysis to an earlier point in the manuscript makes the overall flow better. Therefore, we moved the section right after “Complex members tend to be co-deregulated”. Furthermore, we divided the previous section into two parts: (1) “Epigenetic and transcriptional control cannot fully explain the dysregulation on other chromosomes” where we focus on methylation and TF analysis (page 6-7); and (2) “Post-translational regulation of partner co-abundance” where we focus on co-abundance regulation of co-complex members and the role of post-translational mechanisms (page 7-8).

b) Rather I think the authors could directly look at the transcript levels to assess the degree of impact of transcriptional regulation on the expression changes of complex partners? This would be a powerful analysis better suited for figure 5.

To test to which degree the co-abundance regulation of co-complex members could be explained by transcriptional control, we, first, compared methylation levels of co-complex members between aneuploid and diploid samples. Second, we tested if the transcript-level changes of co-complex members could be explained by the differential activity of TFs encoded on aneuploid chromosomes. Overall, we did not find significant associations both for methylation patterns and TF regulation. We extended the Results section (page 8 – line numbers 5-10), the Materials and methods section (page 17 – line numbers 17-23) including these analyses, and show the results in Figure 3—figure supplement 2C, D.

4) Panel 5B protein ubiquitination is quite nice, but I have not seen any correlation between the number of ubiquitination sites and the propensity to degrade fast. The authors could perhaps reference some literature for this or do their own correlation analysis between turnover rates and number of ubiquitination-sites.

We used the number of ubiquitination sites on a protein as a proxy for its regulation by ubiquitin-mediated degradation as in several studies the number of ubiquitination sites has been suggested to affect the binding of the degradation machinery and its efficiency:

– Lu et al., showed that a larger number of ubiquitins increase the binding affinity between substrate and proteasome, and further increase the dwell time on the proteasome4.

– Dimova et al., showed that ubiquitination of multiple lysine residues is an efficient signal for degradation and forming ubiquitin chains becomes necessary only when the number of lysine residues is restricted in *Xenopus* extracts5.

We now referenced these studies in the Results section (page 8 – line numbers 11-13). Of course, having direct measurements of ubiquitination would be better but in the absence of this data, we believe that using the number of ubiquitination sites as a proxy for regulatory potential through ubiquitination is a reasonable choice.

a) In addition, it would be better to leave out proteins lacking ubiquitination-sites in the database rather than assume they're lacking (and thus give them a 0 value) as this would bias the data heavily towards very abundant proteins, such as the ribosome.

We repeated the analysis by removing proteins that are not covered by the PhosphoSitePlus ubiquitome data. The tendency in which top correlated proteins are significantly related to having a higher number of ubiquitination sites was still observed except for the top negatively correlated proteins in the amplification cases (p = 0.12, Wilcoxon test).

We now revised the figure (Figure 3B – previous Figure 5B) and did the corresponding changes in the Materials and methods section (page 17 – line numbers 30-31).

b) There is some good literature on the E3 ubiquitin ligase (Yanagitani, Juszkiewicz, and Hegde 2017; Nguyen et al., 2017) that targets orphan subunits and ubiquitinates proteins at sites that are usually protected while they're in a complex. Looking at ubiquitination sites in interaction surfaces would be a more mechanistically attractive analysis but probably less feasible?

We downloaded protein interaction interface information from Interactome INSIDER6, and counted the number of ubiquitination sites only in the interfaces. We found some significant associations. For example, top positively correlated co-complex members of deleted proteins are related to a larger number of ubiquitination sites in their interfaces when compared to that of all proteins (see Author response figure 3). It might suggest that ubiquitination sites in the interfaces of top positively correlated co-complex members of deleted proteins will be easily targeted by E3 ligase and degraded (if we consider they will not form complexes with other complex members – or together with the deleted aneuploid protein they could act as a scaffold for the formation of the complex). However, the coverage between the structural and ubiquitome data is very low (e.g. we could not include top negative correlated proteins in the amplification cases as only one pair is covered by both sets) and hence did not feel confident to include this analysis to the manuscript.

**Author response image 3. sa2fig3:** The number of ubiquitylated sites in the interfaces of proteins.